# PREMIER–TACO IS A FEW-SHOT POLICY LEARNER: PRETRAINING MULTITASK REPRESENTATION VIA TEMPORAL ACTION-DRIVEN CONTRASTIVE LOSS

## ABSTRACT

We introduce `Premier-TACO`, a novel multitask feature representation learning methodology aiming to enhance the efficiency of few-shot policy learning in sequential decision-making tasks. `Premier-TACO` pretrains a general feature representation using a small subset of relevant multitask offline datasets, capturing essential environmental dynamics. This representation can then be fine-tuned to specific tasks with few expert demonstrations. Building upon the recent temporal action contrastive learning (TACO) objective, which obtains the state of art performance in visual control tasks, `Premier-TACO` additionally employs a simple yet effective negative example sampling strategy. This key modification ensures computational efficiency and scalability for large-scale multitask offline pretraining. Experimental results from both Deepmind Control Suite and MetaWorld domains underscore the effectiveness of `Premier-TACO` for pretraining visual representation, facilitating efficient few-shot imitation learning of unseen tasks. On the DeepMind Control Suite, `Premier-TACO` achieves an average improvement of 101% in comparison to a carefully implemented Learn-from-scratch baseline, and a 24% improvement compared with the most effective baseline pretraining method. Similarly, on MetaWorld, `Premier-TACO` obtains an average advancement of 74% against Learn-from-scratch and a 40% increase in comparison to the best baseline pretraining method.

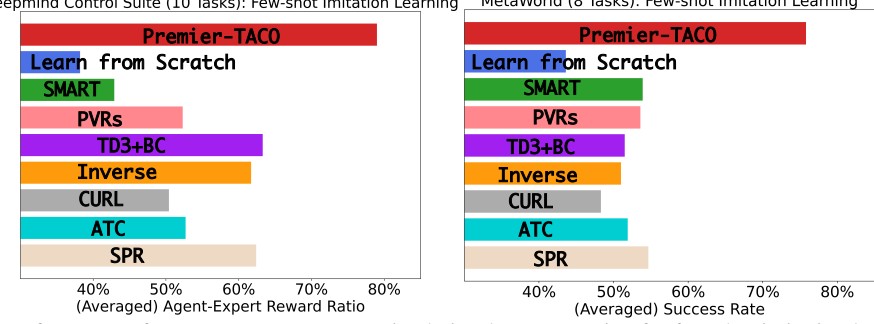

**Figure 1:** Performance of `Premier-TACO` pretrained visual representation for few-shot imitation learning on downstream unseen tasks from both Deepmind Control Suite and MetaWorld.

## 1 INTRODUCTION

In the dynamic and ever-changing world we inhabit, the importance of sequential decision-making (SDM) in machine learning cannot be overstated. Unlike static tasks, sequential decisions reflect the fluidity of real-world scenarios, from robotic manipulations to evolving healthcare treatments. Just as foundation models in language, such as BERT (Devlin et al., 2019) and GPT (Radford et al., 2019; Brown et al., 2020), have revolutionized natural language processing by leveraging vast amounts of textual data to understand linguistic nuances, *pretrained foundation models* hold similar promise for sequential decision-making (SDM). In language, these models capture the essence of syntax, semantics, and context, serving as a robust starting point for a myriad of downstream

tasks. Analogously, in SDM, where decisions are influenced by a complex interplay of past actions, current states, and future possibilities, a pretrained foundation model can provide a rich, generalized understanding of decision sequences. This foundational knowledge, built upon diverse decision-making scenarios, can then be fine-tuned to specific tasks, much like how language models are adapted to specific linguistic tasks.

The following **challenges** are unique to sequential decision-making, setting it apart from existing vision and language pretraining paradigms. **(C1) Data Distribution Shift**: Training data usually consists of specific behavior-policy-generated trajectories. This leads to vastly different data distributions at various stages—pretraining, finetuning, and deployment—resulting in compromised performance (Lee et al., 2021). **(C2) Task Heterogeneity**: Unlike language and vision tasks, which often share semantic features, decision-making tasks vary widely in configurations, transition dynamics, and state and action spaces. This makes it difficult to develop a universally applicable representation. **(C3) Data Quality and Supervision**: Effective representation learning often relies on high-quality data and expert guidance. However, these resources are either absent or too costly to obtain in many real-world decision-making tasks (Brohan et al., 2023; Stooke et al., 2021b). Our **aspirational criteria** for foundation model for sequential decision-making encompass several key features: **(W1) Versatility** that allows the model to generalize across a wide array of tasks, even those not previously encountered, such as new embodiments viewed or observations from novel camera angles; **(W2) Efficiency** in adapting to downstream tasks, requiring minimal data through few-shot learning techniques; **(W3) Robustness** to pretraining data of fluctuating quality, ensuring a resilient foundation; and **(W4) Compatibility** with existing large pretrained models such as Nair et al. (2022).

In light of these challenges and desirables in building foundation models for SDM, our approach to develop foundational models for sequential decision-making focuses on creating a universal and transferable encoder using a reward-free, dynamics based, temporal contrastive pretraining objective. This encoder would be tailored to manage tasks with complex observation spaces, such as visual inputs. By excluding reward signals during the pretraining stage, the model will be better poised to generalize across a broad array of downstream tasks that may have divergent objectives. Leveraging a world-model approach ensures that the encoder learns a compact representation that can capture universal transition dynamics, akin to the laws of physics, thereby making it adaptable for multiple scenarios. This encoder enables the transfer of knowledge to downstream control tasks, even when such tasks were not part of the original pretraining data set.

Existing works apply self-supervised pre-training from rich vision data such as ImageNet (Deng et al., 2009) or Ego4D datasets (Grauman et al., 2022) to build foundation models (Nair et al., 2022; Majumdar et al., 2023; Ma et al., 2023). However, applying these approaches to sequential decision-making tasks is challenging. Specifically, they often overlook control-relevant considerations and suffer from a domain gap between pre-training datasets and downstream visuo-motor tasks. In this paper, rather than focusing on leveraging large computational vision datasets, we propose a novel control-centric objective function for pretraining. Our approach, called `Premier-TACO` (pretraining multitask representation via temporal action-driven contrastive loss), employs a temporal action-driven contrastive loss function for pretraining. Unlike TACO, which treats every data point in the batch as a potential negative example, `Premier-TACO` samples one negative example from a nearby window of the next state, yielding a negative example that is visually similar to the positive one. Consequently, the latent representation must encapsulate control-relevant information to differentiate between the positive and negative examples, rather than depending on irrelevant features such as visual appearance. This simple yet effective negative example sampling strategy incurs zero computational overhead, and through extensive empirical evaluation, we verify with extensive empirical evaluation that `Premier-TACO` works well with smaller batch sizes. Thus `Premier-TACO` can be effortlessly scaled up for multitask offline pretraining.

Below we list our key contributions:

▷ **(1)** We introduce `Premier-TACO`, a new framework designed for the multi-task offline visual representation pretraining of sequential decision-making problems. In particular, we develop a new temporal contrastive learning objective within the `Premier-TACO` framework. Compared with other temporal contrastive learning objectives such as TACO, `Premier-TACO` employs a simple yet efficient negative example sampling strategy, making it computationally feasible for multi-task representation learning.

▷ **(2) [(W1) Versatility (W2) Efficiency]** Through extensive empirical evaluation, we verify the effectiveness of `Premier-TACO`'s pretrained visual representations for few-shot learning on unseen tasks. On MetaWorld (Yu et al., 2019), with 5 expert trajectories, `Premier-TACO` out-performs the best baseline pretraining method by 37%. On Deepmind Control Suite (DMC) (Tassa et al., 2018), using only 20 trajectories, which is considerably fewer demonstrations than (Sun et al., 2023; Majumdar et al., 2023), `Premier-TACO` achieves the best performance across 10 challenging tasks, including the hard Dog and Humanoid tasks. This versatility extends even to unseen embodiments in DMC as well as unseen tasks with unseen camera views in MetaWorld.

▷ **(3) [(W3) Robustness (W4) Compatability]** Furthermore, we demonstrate that `Premier-TACO` is not only resilient to data of lower quality but also compatible with exisiting large pretrained models. In DMC, `Premier-TACO` works well with the pretraining dataset collected randomly. Additionally, we showcase the capability of the temporal contrastive learning objective of `Premier-TACO` to finetune a generalized visual encoder such as R3M (Nair et al., 2022), resulting in an averaged performance enhancement of around 50% across the assessed tasks.

## 2  PRELIMINARY

### 2.1  MULTITASK OFFLINE PRETRAINING

We consider a collection of tasks $\left\{\mathcal{T}_i : (\mathcal{X}, \mathcal{A}_i, \mathcal{P}_i, \mathcal{R}_i, \gamma)\right\}_{i=1}^N$ with the same dimensionality in observation space $\mathcal{X}$. Let $\phi : \mathcal{X} \rightarrow \mathcal{Z}$ be a representation function of the agent's observation, which is either randomly initialized or pre-trained already on a large-scale vision dataset such as ImageNet (Deng et al., 2009) or Ego4D (Grauman et al., 2022). Assuming that the agent is given a multitask offline dataset $\{(x_i, a_i, x_i', r_i)\}$ of a subset of $K$ tasks $\{\mathcal{T}_{n_j}\}_{j=1}^K$. The objective is to pretrain a generalizable state representation $\phi$ or a motor policy $\pi$ so that when facing an unseen downstream task, it could quickly adapt with few expert demonstrations, using the pretrained representation. Below we summarize the pretraining and finetuning setups.

**Pretraining**: The agent get access to a multitask offline dataset, which could be highly suboptimal. The goal is to learn a generalizable shared state representation from pixel inputs.

**Adaptation**: Adapt to unseen downstream task from few expert demonstration with imitation learning.

### 2.2  TACO: TEMPORAL ACTION DRIVEN CONTRASTIVE LEARNING OBJECTIVE

Temporal Action-driven Contrastive Learning (TACO) (Zheng et al., 2023) is a reinforcement learning algorithm proposed for addressing the representation learning problem in visual continuous control. It aims to maximize the mutual information between representations of current states paired with action sequences and representations of the corresponding future states:

$$\mathbb{J}_{\text{TACO}} = \mathcal{I}(Z_{t+K}; [Z_t, U_t, ..., U_{t+K-1}]) \tag{1}$$

Here, $Z_t = \phi(X_t)$ and $U_t = \psi(A_t)$ represent latent state and action variables. Theoretically, it could be shown that maximization of this mutual information objective lead to state and action representations that are capable of representing the optimal value functions. Empirically, TACO estimate the lower bound of the mutual information objective by the InfoNCE loss, and it achieves the state of art performance for both online and offline visual continuous control, demonstrating the effectiveness of temporal contrastive learning for representation learning in sequential decision making problems.

## 3  METHOD

We introduce `Premier-TACO`, a generalized pre-training approach specifically formulated to tackle the multi-task pre-training problem, enhancing sample efficiency and generalization ability for downstream tasks. Building upon the success of temporal contrastive loss, exemplified by `TACO` (Zheng et al., 2023), in acquiring latent state representations that encapsulate individual task dynamics, our aim is to foster representation learning that effectively captures the intrinsic dynamics spanning a diverse set of tasks found in offline datasets. Our overarching objective is to ensure that these learned representations exhibit the versatility to generalize across unseen tasks that share the underlying dynamic structures.

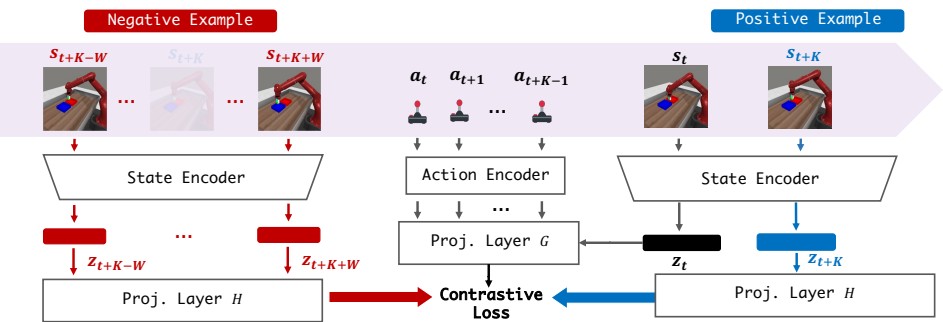

**Figure 2:** An illustration of `Premier-TACO` contrastive loss design. The two 'State Encoder's are identical, as are the two 'Proj. Layer $H$'s. One negative example is sampled from the neighbors of framework $s_{t+K}$.

Nevertheless, when adapted for multitask offline pre-training, the online learning objective of TACO (Zheng et al., 2023) poses a notable challenge. Specifically, TACO's mechanism, which utilizes the InfoNCE (van den Oord et al., 2019) loss, categorizes all subsequent states $s_{t+k}$ in the batch as negative examples. While this methodology has proven effective in single-task reinforcement learning scenarios, it encounters difficulties when extended to a multitask context. During multitask offline pretraining, image observations within a batch can come from different tasks with vastly different visual appearances, rendering the contrastive InfoNCE loss significantly less effective.

**Offline Pretraining Objective.** We propose a straight-forward yet highly effective mechanism for selecting challenging negative examples. Instead of treating all the remaining examples in the batch as negatives, `Premier-TACO` selects the negative example from a window centered at state $s_{t+k}$ within the same episode.

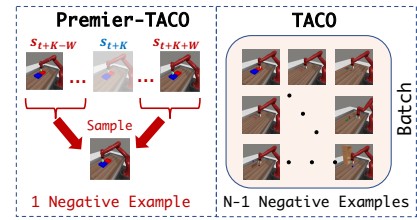

This approach is both computationally efficient and more statistically powerful due to negative examples which are challenging to distinguish from similar positive examples forcing the model capture temporal dynamics differentiating between positive and negative examples. Specifically, given a batch of state and action sequence transitions

**Figure 3:** Difference between `Premier-TACO` and TACO for sampling negative examples

$\{(s_t^{(i)}, [a_t^{(i)}, ..., a_{t+K-1}^{(i)}], s_{t+K}^{(i)})\}_{i=1}^N$ , let $z_t^{(i)} = \phi(s_t^{(i)})$, $u_t^{(i)} = \psi(a_t^{(i)})$ be latent state and latent action embeddings respectively. Furthermore, let $\widetilde{s_{t+K}^{(i)}}$ be a negative example uniformly sampled from the window of size $W$ centered at $s_{t+K}$: $(s_{t+K-W}, ..., s_{t+K-1}, s_{t+K+1}, ..., s_{t+K+W})$ with $\widetilde{z_t^{(i)}} = \phi(\widetilde{s_t^{(i)}})$ a negative latent state. Given these, define $g_t^{(i)} = G_\theta(z_t^{(i)}, u_t^{(i)}, ..., u_{t+K-1}^{(i)})$, $\widetilde{h_t^{(i)}} = H_\theta(\widetilde{z_{t+K}^{(i)}})$, and $h_t^{(i)} = H_\theta(z_{t+K}^{(i)})$ as embeddings of future predicted and actual latent states. We optimize:

$$\mathcal{J}_{\texttt{Premier-TACO}}(\phi, \psi, G_\theta, H_\theta) = -\frac{1}{N}\sum_{i=1}^{N}\log\frac{{g_t^{(i)}}^\top h_{t+K}^{(i)}}{{g_t^{(i)}}^\top h_{t+K}^{(i)} + \widetilde{g_t^{(i)}}^\top h_{t+K}^{(i)}}. \quad (2)$$

**Pretraining of `Premier-TACO`.** For representation pretraining, we construct an offline dataset that includes control trajectories generated by behavioral policies across a diverse set of tasks. Throughout the pretraining phase, we employ the `Premier-TACO` learning objective to update a randomly initialized shallow ConvNet encoder. Unlike prior representation pretraining approach (Sun et al., 2023; Nair et al., 2022; Majumdar et al., 2023), we firmly believe that the `Premier-TACO` approach does not impose stringent requirements on the network structure of the representation encoder. Concerning the selection of task data for pretraining, our strategy intentionally covers various embodiments for Deepmind Control Suite and encompasses a wide range of motion patterns and interactions involving robotic arms interacting with different objects for MetaWorld. Additionally, we prioritize choosing simpler tasks for pretraining to demonstrate the pretrained model's effective generalization to more challenging unseen tasks.

**Few-shot Generalization.** After pretraining the representation encoder, we leverage our pretrained model $\Phi$ to learn policies for downstream tasks. To learn the policy $\pi$ with the state representation $\Phi(s_t)$ as inputs, we use behavior cloning (BC) with a few expert demonstrations. For different control domains, we employ significantly fewer demonstrations for unseen tasks than what is typically used in other baselines. This underscores the substantial advantages of `Premier-TACO` in few-shot generalization. More details about the experiments on downstream tasks will be provided in Section 4.

## 4 EXPERIMENT

In our empirical evaluations, we consider two benchmarks, Deepmind Control Suite (Tassa et al., 2018) for locomotion control as well as MetaWorld (Yu et al., 2019) for robotic manipulation tasks.

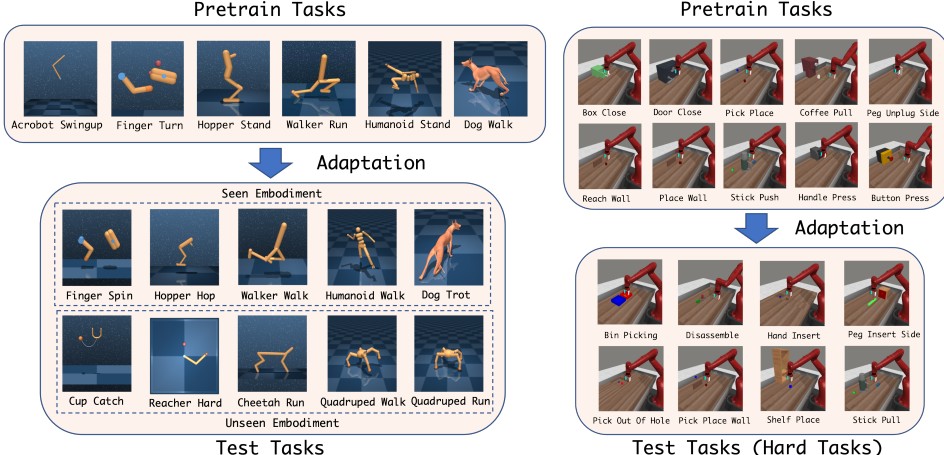

**Figure 4:** Pretrain and Test Tasks split for Deepmind Control Suite and MetaWorld. The left figures are Deepmind Control Suite tasks and the right figures MetaWorld tasks.

**Deepmind Control Suite (DMC) (Tassa et al., 2018)**: We consider a selection of 16 challenging tasks from Deepmind Control Suite. Note that compared with prior works such as Majumdar et al. (2023); Sun et al. (2023), we consider much harder tasks, including ones from the humanoid and dog domains, which feature intricate kinematics, skinning weights and collision geometry. For pretraining, we select six tasks (**DMC-6**), including Acrobot Swingup, Finger Turn Hard, Hopper Stand, Walker Run, Humanoid Walk, and Dog Stand. We generate an exploratory dataset for each task by sampling trajectories generated in exploratory stages of a DrQ-v2 (Yarats et al., 2022) learning agent. In particular, we sample 1000 trajectories from the online replay buffer of DrQ-v2 once it reaches the convergence performance. This ensures the diversity of the pretraining data, but in practice, such a high-quality dataset could be hard to obtain. So, later in the experiments, we will also relax this assumption and consider pretrained trajectories that are sampled from uniformly random actions.
**MetaWorld (Yu et al., 2019)**: We select a set of 10 tasks for pretraining, which encompasses a variety of motion patterns of the Sawyer robotic arm and interaction with different objects. To collect an exploratory dataset for pretraining, we execute the scripted policy with Gaussian noise of a standard deviation of 0.3 added to the action. By adding such a noise, the success rate of collected policies on average is only around 20% across ten pretrained tasks.

**Baselines.** We compare `Premier-TACO` with the following representation pretraining baselines:

▷ Learn from Scratch: Behavior Cloning with randomly initialized shallow ConvNet encoder. Different from Nair et al. (2022); Majumdar et al. (2023), which use a randomly initialized ResNet for evaluation, we find that using a shallow network with an input image size of $84 \times 84$ on both Deepmind Control Suite and MetaWorld yields superior performance. Additionally, we also include data augmentation into behavior cloning following Hansen et al. (2022a).

▷ Policy Pretraining: We first train a multitask policy by TD3+BC (Fujimoto & Gu, 2021) on the pretraining dataset. While numerous alternative offline RL algorithms exist, we choose TD3+BC

| DMControl | | | | | Models | | | | | |
|---|---|---|---|---|---|---|---|---|---|---|
| **Tasks** | LfS | SMART | Best PVRs | TD3+BC | Inverse | CURL | ATC | SPR | TACO | **Premier-TACO** |
| **Seen Embodiments** | | | | | | | | | | |
| Finger Spin | $34.8\pm3.4$ | $44.2\pm8.2$ | $38.4\pm9.3$ | $68.8\pm7.1$ | $33.4\pm8.4$ | $35.1\pm9.6$ | $51.1\pm9.4$ | $55.9\pm6.2$ | $28.4\pm9.7$ | $\mathbf{75.2\pm0.6}$ |
| Hopper Hop | $8.0\pm1.3$ | $14.2\pm3.9$ | $23.2\pm4.9$ | $49.1\pm4.3$ | $48.3\pm5.2$ | $28.7\pm5.2$ | $34.9\pm3.9$ | $52.3\pm7.8$ | $21.4\pm3.4$ | $\mathbf{75.3\pm4.6}$ |
| Walker Walk | $30.4\pm2.9$ | $54.1\pm5.2$ | $32.6\pm8.7$ | $65.8\pm2.0$ | $64.4\pm5.6$ | $37.3\pm7.9$ | $44.6\pm5.0$ | $72.9\pm1.5$ | $30.6\pm6.1$ | $\mathbf{88.0\pm0.8}$ |
| Humanoid Walk | $15.1\pm1.3$ | $18.4\pm3.9$ | $30.1\pm7.5$ | $34.9\pm8.5$ | $41.9\pm8.4$ | $19.4\pm2.8$ | $35.1\pm3.1$ | $30.1\pm6.2$ | $29.1\pm8.1$ | $\mathbf{51.4\pm4.9}$ |
| Dog Trot | $52.7\pm3.5$ | $59.7\pm5.2$ | $73.5\pm6.4$ | $82.3\pm4.4$ | $85.3\pm2.1$ | $71.9\pm2.2$ | $84.3\pm0.5$ | $79.9\pm3.8$ | $80.1\pm4.1$ | $\mathbf{93.9\pm5.4}$ |
| **Unseen Embodiments** | | | | | | | | | | |
| Cup Catch | $56.8\pm5.6$ | $66.8\pm6.2$ | $93.7\pm1.8$ | $97.1\pm1.7$ | $96.7\pm2.6$ | $96.7\pm2.6$ | $96.2\pm1.4$ | $96.9\pm3.1$ | $88.7\pm3.2$ | $\mathbf{98.9\pm0.1}$ |
| Reacher Hard | $34.6\pm4.1$ | $52.1\pm3.8$ | $64.9\pm5.8$ | $59.6\pm9.9$ | $61.7\pm4.6$ | $50.4\pm4.6$ | $56.9\pm9.8$ | $62.5\pm7.8$ | $58.3\pm6.4$ | $\mathbf{81.3\pm1.8}$ |
| Cheetah Run | $25.1\pm2.9$ | $41.1\pm7.2$ | $39.5\pm9.7$ | $50.9\pm2.6$ | $51.5\pm5.5$ | $36.8\pm5.4$ | $30.1\pm1.0$ | $40.2\pm9.6$ | $23.2\pm3.3$ | $\mathbf{65.7\pm1.1}$ |
| Quadruped Walk | $61.1\pm5.7$ | $45.4\pm4.3$ | $63.2\pm4.0$ | $76.6\pm7.4$ | $82.4\pm6.7$ | $72.8\pm8.9$ | $81.9\pm5.6$ | $65.6\pm4.0$ | $63.9\pm9.3$ | $\mathbf{83.2\pm5.7}$ |
| Quadruped Run | $45.0\pm2.9$ | $27.9\pm5.3$ | $64.0\pm2.4$ | $48.2\pm5.2$ | $52.1\pm1.8$ | $55.1\pm5.4$ | $2.6\pm3.6$ | $68.2\pm3.2$ | $50.8\pm5.7$ | $\mathbf{76.8\pm7.5}$ |
| **Mean Performance** | 38.2 | 42.9 | 52.3 | 63.3 | 61.7 | 50.4 | 52.7 | 62.4 | 47.5 | **79.0** |

**Table 1: [(W1) Versatility (W2) Efficiency] Few-shot Behavior Cloning (BC) for unseen task of DMC.** Performance (Agent Reward / Expert Reward) of baselines and `Premier-TACO` on 10 unseen tasks on Deepmind Control Suite. **Bold** numbers indicate the best results. Agent Policies are evaluated every 1000 gradient steps for a total of 100000 gradient steps and we report the average performance over the 3 best epochs over the course of learning. `Premier-TACO` outperforms all the baselines, showcasing its superior efficacy in generalizing to unseen tasks with seen or **unseen embodiments**.

| MetaWorld | | | | | Models | | | | | |
|---|---|---|---|---|---|---|---|---|---|---|
| **Unseen Tasks** | LfS | SMART | Best PVRs | TD3+BC | Inverse | CURL | ATC | SPR | TACO | **Premier-TACO** |
| Bin Picking | $62.5\pm12.5$ | $71.3\pm9.6$ | $60.2\pm4.3$ | $50.6\pm3.7$ | $55.0\pm7.9$ | $45.6\pm5.6$ | $55.6\pm7.8$ | $67.9\pm6.4$ | $67.3\pm7.5$ | $\mathbf{78.5\pm7.2}$ |
| Disassemble | $56.3\pm6.5$ | $52.9\pm4.5$ | $70.4\pm8.9$ | $56.9\pm11.5$ | $53.8\pm8.1$ | $66.2\pm8.3$ | $45.6\pm9.8$ | $48.8\pm5.4$ | $51.3\pm10.8$ | $\mathbf{86.7\pm8.9}$ |
| Hand Insert | $34.7\pm7.5$ | $34.1\pm5.2$ | $35.5\pm2.3$ | $46.2\pm5.2$ | $50.0\pm3.5$ | $49.4\pm7.6$ | $51.2\pm1.3$ | $52.4\pm5.2$ | $56.8\pm4.2$ | $\mathbf{75.0\pm7.1}$ |
| Peg Insert Side | $28.7\pm2.0$ | $20.9\pm3.6$ | $48.2\pm3.6$ | $30.0\pm6.1$ | $33.1\pm6.2$ | $28.1\pm3.7$ | $31.8\pm4.8$ | $39.2\pm7.4$ | $36.3\pm4.5$ | $\mathbf{62.7\pm4.7}$ |
| Pick Out Of Hole | $53.7\pm6.7$ | $65.9\pm7.8$ | $66.3\pm7.2$ | $46.9\pm7.4$ | $50.6\pm5.1$ | $43.1\pm6.2$ | $54.4\pm8.5$ | $55.3\pm6.8$ | $52.9\pm7.3$ | $\mathbf{72.7\pm7.25}$ |
| Pick Place Wall | $40.5\pm4.5$ | $62.8\pm5.9$ | $63.2\pm9.8$ | $63.8\pm12.4$ | $71.3\pm11.3$ | $73.8\pm11.9$ | $68.7\pm5.5$ | $72.3\pm7.5$ | $37.8\pm8.5$ | $\mathbf{80.2\pm8.2}$ |
| Shelf Place | $26.3\pm4.1$ | $57.9\pm4.5$ | $32.4\pm6.5$ | $45.0\pm7.7$ | $36.9\pm6.7$ | $35.0\pm10.8$ | $35.6\pm10.7$ | $38.0\pm6.5$ | $25.8\pm5.0$ | $\mathbf{70.4\pm8.1}$ |
| Stick Pull | $46.3\pm7.2$ | $65.8\pm8.2$ | $52.4\pm5.6$ | $72.3\pm11.9$ | $57.5\pm9.5$ | $43.1\pm15.2$ | $72.5\pm8.9$ | $68.5\pm9.4$ | $52.0\pm10.5$ | $\mathbf{80.0\pm8.1}$ |
| **Mean** | 43.6 | 53.9 | 53.6 | 51.5 | 51.0 | 48.3 | 51.9 | 55.3 | 47.5 | **75.8** |

**Table 2: [(W1) Versatility (W2) Efficiency] Five-shot Behavior Cloning (BC) for unseen task of MetaWorld.** Success rate of `Premier-TACO` and baselines across 8 hard unseen tasks on MetaWorld. Results are aggregated over 4 random seeds. **Bold** numbers indicate the best results.

as a representative due to its simplicity and great empirical performance. After pretraining, we take the pretrained ConvNet encoder and drop the policy MLP layers.

▷ Pretrained Visual Representations (PVRs): We evaluate the state-of-the-art frozen pretrained visual representations including PVR (Parisi et al., 2022), MVP (Xiao et al., 2022), R3M (Nair et al., 2022) and VC-1 (Majumdar et al., 2023), and report the best performance of these PVRs models for each task.

▷ Control Transformer: SMART (Sun et al., 2023) is a self-supervised representation pretraining framework which utilizes a maksed prediction objective for pretraining representation under Decision Transformer architecture, and then use the pretrained representation to learn policies for downstream tasks.

▷ Inverse Dynamics Model: We pretrain an inverse dynamics model to predict actions and use the pretrained representation for downstream task.

▷ Contrastive/Self-supervised Learning Objectives: CURL (Laskin et al., 2020), ATC (Stooke et al., 2021a), SPR (Schwarzer et al., 2021a;b). CURL and ATC are two approaches that apply contrastive learning into sequential decision making problems. While CURL treats augmented states as positive pairs, it neglects the temporal dependency of MDP. In comparison, ATC takes the temporal structure into consideration. The positive example of ATC is an augmented view of a temporally nearby state. SPR applies BYOL objecive (Grill et al., 2020) into sequential decision making problems by pretraining state representations that are self-predictive of future states.

**Pretrained feature representation by `Premier-TACO` facilitates effective few-shot adaptation to unseen tasks.** We measure the performance of pretrained visual representation for few-shot imitation learning of unseen downstream tasks in both DMC and MetaWorld. In particular, for DMC, we use **20 expert trajectories** for imitation learning except for the two hardest tasks, Humanoid Walk and Dog Trot, for which we use 100 trajectories instead. Note that we only use $\frac{1}{5}$ of the number of expert trajectories used in Majumdar et al. (2023) and $\frac{1}{10}$ of those used in Sun et al. (2023).

We record the performance of the agent by calculating the ratio of $\frac{\text{Agent Reward}}{\text{Expert Reward}}$, where Expert Reward is the episode reward of the expert policy used to collect demonstration trajectories. For MetaWorld, we use **5 expert trajectories** for all eight downstream tasks, and we use task success rate as the performance metric. In Table 1 and Table 2, we present the results for Deepmind Control Suite and MetaWorld, respectively. As shown here, pretrained representation of `Premier-TACO` significantly improves the few-shot imitation learning performance compared with Learn-from-scratch, with a **101%** improvement on Deepmind Control Suite and **74%** improvement on MetaWorld, respectively. Moreover, it also outperforms all the baselines across all tasks by a large margin.

**`Premier-TACO` pre-trained representation enables knowledge sharing across different embodiments.** Ideally, a resilient and generalizable state feature representation ought not only to encapsulate universally applicable features for a given embodiment across a variety of tasks, but also to exhibit the capability to generalize across distinct embodiments. Here, we evaluate the few-shot behavior cloning performance of `Premier-TACO` pre-trained encoder from **DMC-6** on four tasks featuring unseen embodiments: Cup Catch, Cheetah Run, and Quadruped Walk. In comparison to Learn-from-scratch, as shown in Figure 5, `Premier-TACO` pre-trained representation realizes an **82%** performance gain, demonstrating the robust generalizability of our pre-trained feature representations.

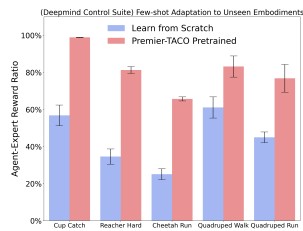

**Figure 5:** **[(W1) Versatility]** DMControl: Generalization of `Premier-TACO` pre-trained visual representation to unseen embodiments.

**`Premier-TACO` Pretrained Representation is also generalizable to unseen tasks with camera views.** Beyond generalizing to unseen embodiments, an ideal robust visual representation should possess the capacity to adapt to unfamiliar tasks under novel camera views. In Figure 6, we evaluate the five-shot learning performance of our model on four previously unseen tasks in MetaWorld with a new view. In particular, during pretraining, the data from MetaWorld are generated using the same view as employed in (Hansen et al., 2022b; Seo et al., 2022). Then for downstream policy learning, the agent is given five expert trajectories under a different corner camera view, as depicted in the figure. Notably, `Premier-TACO` also achieves a substantial performance enhancement, thereby underscoring the robust generalizability of our pretrained visual representation.

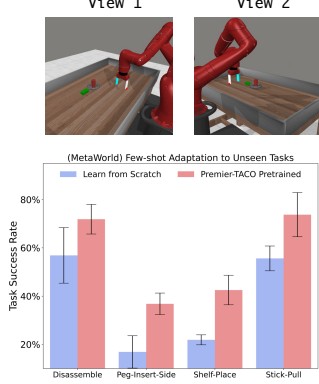

**Figure 6:** **[(W1) Versatility]** MetaWorld: Few-shot adaptation to unseen tasks from an unseen camera view.

**`Premier-TACO` Pre-trained Representation is resilient to low-quality data.** We evaluate the resilience of `Premier-TACO` by employing randomly collected trajectory data from Deepmind Control Suite for pretraining and compare it with `Premier-TACO` representations pretrained using an exploratory dataset and the learn-from-scratch approach. As illustrated in Figure 7, across all downstream tasks, even when using randomly pretrained data, the `Premier-TACO` pretrained model still maintains a significant advantage over learning-from-scratch. When compared with representations pretrained using exploratory data, there are only small disparities in a few individual tasks, while they remain comparable in most other tasks. This strongly indicates the robustness of `Premier-TACO` to low-quality data. Even without the use of expert control data, our method is capable of extracting valuable information.

**Pretrained visual encoder finetuning with `Premier-TACO`.** In addition to evaluating our pretrained representations across various downstream scenarios, we also conducted fine-tuning on pretrained visual representations using in-domain control trajectories following `Premier-TACO` framework. Importantly, our findings deviate from the observations made in prior works like (Hansen et al., 2022a) and (Majumdar et al., 2023), where fine-tuning of R3M (Nair et al., 2022) on in-domain demonstration data using the task-centric behavior cloning objective, resulted in performance degradation. We speculate that two main factors contribute to this phenomenon. First, a domain gap exists

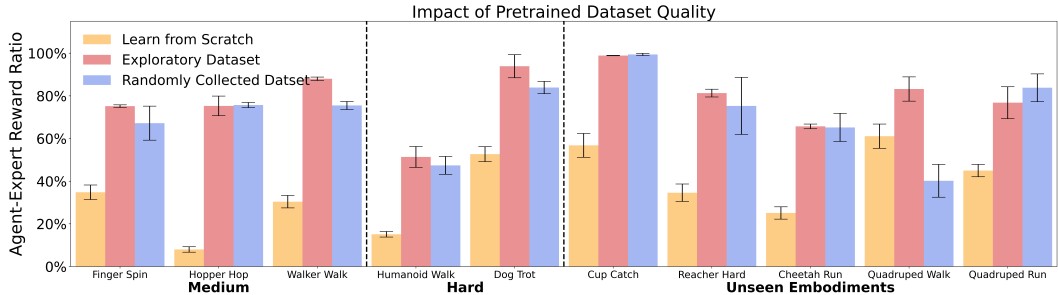

Figure 7: **[(W3) Robustness]** `Premier-TACO` pretrained with exploratory dataset vs. `Premier-TACO` pretrained with randomly collected dataset

between out-of-domain pretraining data and in-domain fine-tuning data. Second, fine-tuning with few-shot learning can lead to overfitting for large pretrained models.

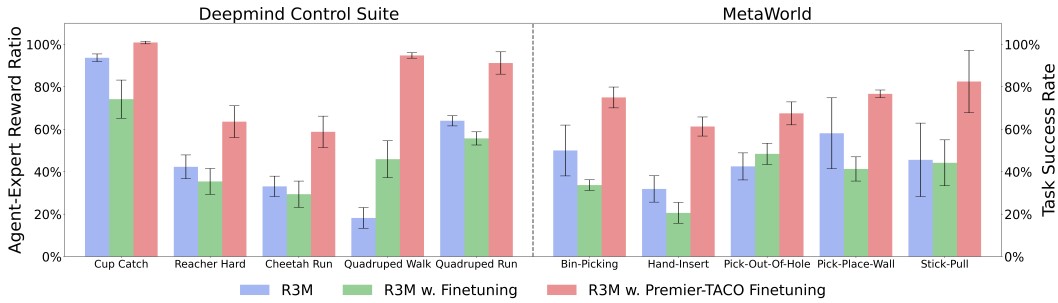

Figure 8: **[(W4) Compatibility]** Finetune R3M (Nair et al., 2022), a generalized Pretrained Visual Encoder with `Premier-TACO` learning objective vs. R3M with in-domain finetuning in Deepmind Control Suite and MetaWorld.

To further validate the effectiveness of our `Premier-TACO` approach, we compared the results of R3M with no fine-tuning, in-domain fine-tuning (Hansen et al., 2022a), and fine-tuning using our method on selected Deepmind Control Suite and MetaWorld pretraining tasks. Figure 8 unequivocally demonstrate that direct fine-tuning on in-domain tasks leads to a performance decline across multiple tasks. However, leveraging the `Premier-TACO` learning objective for fine-tuning substantially enhances the performance of R3M. This not only underscores the role of our method in bridging the domain gap and capturing essential control features but also highlights its robust generalization capabilities. Furthermore, these findings strongly suggest that our `Premier-TACO` approach is highly adaptable to a wide range of multi-task pretraining scenarios, irrespective of the model's size or the size of the pretrained data.

This implies the promising potential to significantly improve the performance of existing pretrained models across diverse domains. The full results of finetuning on all 18 tasks including Deepmind Control Suite and MetaWorld are in Appendix B.1.

**Ablation Study - Batch Size**: Compared with TACO, the negative example sampling strategy employed in `Premier-TACO` allows us to sample harder negative examples within the same episode as the positive example. We expect `Premier-TACO` to work much better with small batch sizes, compared with TACO where the negative examples from a given batch could be coming from various tasks and thus the batch size required would scale up linearly with the number of pretraining tasks. In ours previous experimental results, `Premier-TACO` is pretrained with a batch size of 4096, a standard batch size used in contrastive learning literature. Here, to empirically verify the effects of different choices of the pretraining batch size, we

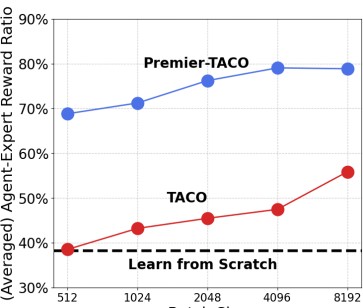

Figure 9: (Updated) Averaged performance of `Premier-TACO` vs. TACO on 10 Deepmind Control Suite Tasks across different batch sizes

train `Premier-TACO` and TACO with different batch sizes and compare their few-shot imitation learning performance.

Figure 9 displays the average performance of few-shot imitation learning across all ten tasks in the DeepMind Control Suite. As depicted in the figure, our model significantly outperform TACO across all batch sizes tested in the experiments, and exhibits performance saturation beyond a batch size of 4096. This observation substantiate that the negative example sampling strategy employed by `Premier-TACO` is indeed the key for the success of multitask offline pretraining.

**Ablation Study - Window Size**: In `Premier-TACO`, the window size $W$ determines the hardness of the negative example. A smaller window size results in negative examples that are more challenging to distinguish from positive examples, though they may become excessively difficult to differentiate in the latent space. Conversely, a larger window size makes distinguishing relatively straightforward, thereby mitigating the impacts of negative sampling. In preceding experiments, a consistent window size of 5 was applied across all trials on both the DeepMind Control Suite and MetaWorld. Here we empirically evaluate the effects of varying window sizes on the average performance of our model across ten DeepMind Control Tasks, as depicted in Figure X. Notably, our observations reveal that performance is comparable when the window size is set to 3, 5, or 7, whereas excessively small ($W = 1$) or large ($W = 9$) window sizes lead to worse performance.

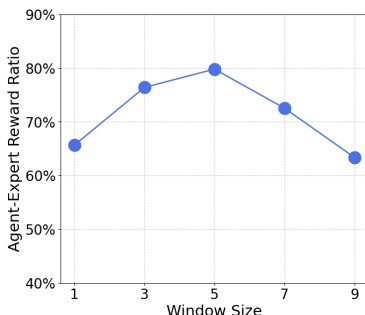

**Figure 10:** Averaged performance of `Premier-TACO` on 10 Deepmind Control Suite Tasks across different window sizes

## 5 RELATED WORK

Existing works, including R3M (Nair et al., 2022), VIP (Ma et al., 2023), MVP (Xiao et al., 2022), PIE-G (Yuan et al., 2022), and VC-1 (Majumdar et al., 2023), focus on self-supervised pre-training for building foundation models but struggle with the domain gap in sequential decision-making tasks. Recent studies, such as one by Hansen et al. (2022a), indicate that models trained from scratch often outperform pre-trained representations. Approaches like SMART (Sun et al., 2023) and DualMind (Wei et al., 2023) offer control-centric pre-training, but at the cost of extensive fine-tuning or task sets. Contrastive learning techniques like CURL (Laskin et al., 2020), CPC (Henaff, 2020), ST-DIM (Anand et al., 2019), and ATC (Stooke et al., 2021a) have succeeded in visual RL, but mainly focus on high-level features and temporal dynamics without a holistic consideration of state-action interactions, a gap partially filled by TACO (Zheng et al., 2023). Our work builds upon these efforts but eliminates the need for extensive task sets and fine-tuning, efficiently capturing control-relevant features. This positions our method as a distinct advancement over DRIML (Mazoure et al., 2020) and Homer (Misra et al., 2019), which require more computational or empirical resources.

A detailed discussion of related work is in Appendix A.

## 6 CONCLUSION

This paper introduces `Premier-TACO`, a robust and highly generalizable representation pretraining framework for few-shot policy learning. We propose a temporal contrastive learning objective that excels in multi-task representation learning during the pretraining phase, thanks to its efficient negative example sampling strategy. Extensive empirical evaluations spanning diverse domains and tasks underscore the remarkable effectiveness and adaptability of `Premier-TACO`'s pre-trained visual representations to unseen tasks, even when confronted with unseen embodiments, different views, and data imperfections. Furthermore, we demonstrate the versatility of `Premier-TACO` by showcasing its ability to fine-tune large pretrained visual representations like R3M (Nair et al., 2022) with domain-specific data, underscoring its potential for broader applications. In our future research agenda, we plan to explore the use of Large Language Models (LLMs) for representation pretraining in policy learning and investigate the applicability of our approach in a wider range of real-world robotics applications, including pretraining visual representations with `Premier-TACO` on real robot datasets such as RoboNet (Dasari et al., 2020) or Bridge-v2 (Walke et al., 2023) datasets.

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

## A    DETAILED DISCUSSION OF RELATED WORK

**Pretraining Visual Representations.** Existing works apply self-supervised pre-training from rich vision data to build foundation models. However, applying this approach to sequential decision-making tasks is challenging. Recent works have explored large-scale pre-training with offline data in the context of reinforcement learning. Efforts such as R3M (Nair et al., 2022), VIP (Ma et al., 2023), MVP (Xiao et al., 2022), PIE-G (Yuan et al., 2022), and VC-1 (Majumdar et al., 2023) highlight this direction. However, there's a notable gap between the datasets used for pre-training and the actual downstream tasks. In fact, a recent study (Hansen et al., 2022a) found that models trained from scratch can often perform better than those using pre-trained representations, suggesting the limitation of these approachs. It's important to acknowledge that these pre-trained representations are not control-relevant, and they lack explicit learning of a latent world model. In contrast to these prior approaches, our pretrained representations learn to capture the control-relevant features with an effective temporal contrastive learning objective.

For control tasks, several pretraining frameworks have emerged to model state-action interactions from high-dimensional observations by leveraging causal attention mechanisms. SMART (Sun et al., 2023) introduces a self-supervised and control-centric objective to train transformer-based models for multitask decision-making, although it requires additional fine-tuning with large number of demonstrations during downstream time. As an improvement, DualMind (Wei et al., 2023) pretrains representations using 45 tasks for general-purpose decision-making without task-specific fine-tuning. Besides, some methods (Sekar et al., 2020; Mendonca et al., 2021; Yarats et al., 2021; Sun et al., 2022) first learn a general representation by exploring the environment online, and then use this representation to train the policy on downstream tasks. In comparison, our approach is notably more efficient and doesn't require training with such an extensive task set. Nevertheless, we provide empirical evidence demonstrating that our method can effectively handle multi-task pretraining.

**Contrastive Representation for Visual RL** Contrastive learning is a self-supervised technique that leverages similarity constraints between data to learn effective representations (embeddings), and it has demonstrated remarkable success across various domains. In the context of visual reinforcement learning (RL), contrastive learning plays a pivotal role in training robust state representations from raw visual inputs, thereby enhancing sample efficiency. CURL (Laskin et al., 2020) extracts high-level features by utilizing InfoNCE(van den Oord et al., 2019) to maximize agreement between augmented observations, although it does not explicitly consider temporal relationships between states. Several approaches, such as CPC (Henaff, 2020), ST-DIM (Anand et al., 2019), and ATC (Stooke et al., 2021a) , introduce temporal dynamics into the contrastive loss. They do so by maximizing mutual information between states with short temporal intervals, facilitating the capture of temporal dependencies. DRIML (Mazoure et al., 2020) proposes a policy-dependent auxiliary objective that enhances agreement between representations of consecutive states, specifically considering the first action of the action sequence. Recent advancements by Kim et al. (2022); Zhang et al. (2021) incorporate actions into the contrastive loss, emphasizing behavioral similarity. TACO (Zheng et al., 2023) takes a step further by learning both state and action representations. It optimizes the mutual information between the representations of current states paired with action sequences and the representations of corresponding future states. In our approach, we build upon the efficient extension of TACO, harnessing the full potential of state and action representations for downstream tasks. On the theory side, the Homer algorithm (Misra et al., 2019) uses a binary temporal contrastive objective reminiscent of the approach used here, which differs by abstracting actions as well states, using an ancillary embedding, removing leveling from the construction, and of course extensive empirical validation.

**Hard Negative Sampling Strategy in Contrastive Learning** Our proposed negative example sampling strategy in `Premier-TACO` is closely related to hard negative example mining in the literature of self-supervised learning as well as other areas of machine learning. Hard negative mining is indeed used in a variety of tasks, such as facial recognition (Wan et al., 2016), object detection (Shrivastava et al., 2016), tracking (Nam & Han, 2016), and image-text retrieval (Pang et al., 2019; Li et al., 2021), by introducing negative examples that are more difficult than randomly chosen ones to improve the performance of models. Within the regime of self-supervised learning, different negative example sampling strategies have also been discussed both empirically and theoretically to improve the quality of pretrained representation. In particular, Robinson et al. (2021) modifies the original NCE objective by developing a distribution over negative examples, which prioritizes

pairs with currently similar representations. Kalantidis et al. (2020) suggests to mix hard negative examples within the latent space. Ma et al. (2021) introduce a method to actively sample uncertain negatives by calculating the gradients of the loss function relative to the model's most confident predictions. Furthermore, Tabassum et al. (2022) samples negatives that combine the objectives of identifying model-uncertain negatives, selecting negatives close to the anchor point in the latent embedding space, and ensuring representativeness within the sample population.

While our proposed approach bears some resemblance to existing negative sampling strategies in contrastive learning literature, we are dealing with unique challenges in sequential decision making, such as data distribution shift, task heterogeneity, and data qualities, as discussed in the introduction. Building on top of the work of TACO (Zheng et al., 2023), which is specifically designed to capture the control-relevant information in latent representation, `Premier-TACO` introduces a straightforward yet effective negative sampling strategy. Tailored toward multitask representation pretraining, this strategy involves sampling one negative example from a window centered around the anchor point, which is both computationally efficient and demonstrates superior performance in few-shot adaptation.

## B  EXPERIMENTS

### B.1  FINETUNING

Comparisons among R3M (Nair et al., 2022), R3M with in-domain finetuning (Hansen et al., 2022a) and R3M finetuned with `Premier-TACO` in Deepmind Control Suite and MetaWorld are presented in Figure 12 and 11.

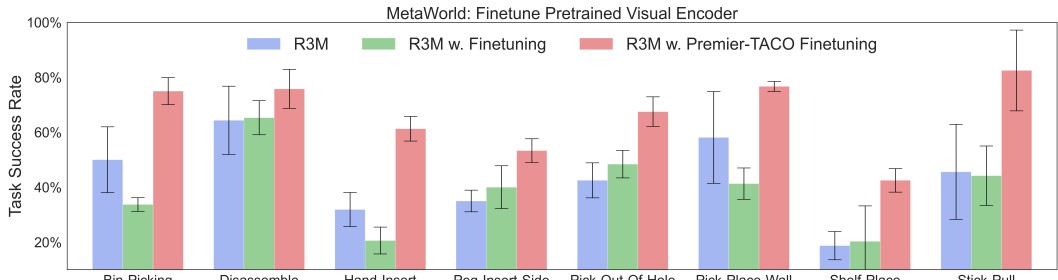

**Figure 11: [(W4) Compatibility]** Finetune R3M (Nair et al., 2022), a generalized Pretrained Visual Encoder with `Premier-TACO` learning objective vs. R3M with in-domain finetuning in Deepmind Control Suite and MetaWorld.

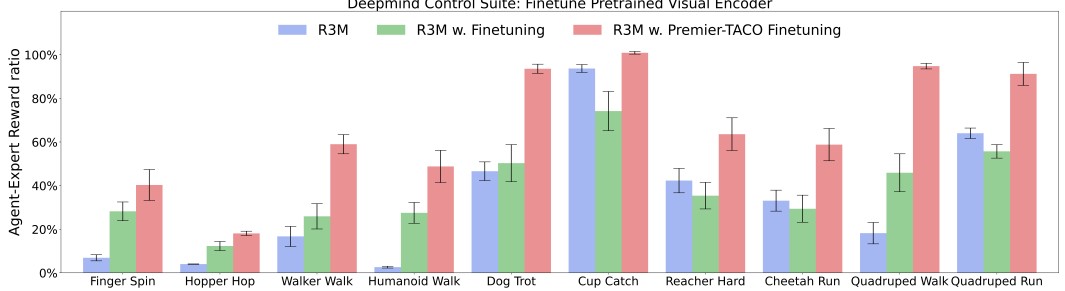

**Figure 12: [(W4) Compatibility]** Finetune R3M (Nair et al., 2022), a generalized Pretrained Visual Encoder with `Premier-TACO` learning objective vs. R3M with in-domain finetuning in Deepmind Control Suite and MetaWorld.

### B.2  PRETRAINED VISUAL REPRESENTATIONS

Here, we provide the full results for all pretrained visual encoders across all 18 tasks on Deepmind Control Suite and MetaWorld.

| DMControl | Pretrained Visual Models | | | | MetaWorld | Pretrained Visual Models | | | |
| | PVR | MVP | R3M | VC-1 | | PVR | MVP | R3M | VC-1 |
| --- | --- | --- | --- | --- | --- | --- | --- | --- | --- |
| Finger Spin | $11.5 \pm 6.0$ | $5.4 \pm 7.1$ | $6.9 \pm 1.4$ | $38.4 \pm 9.3$ | Bin Picking | $45.6 \pm 5.6$ | $46.1 \pm 3.1$ | $50.0 \pm 12.0$ | $60.2 \pm 4.3$ |
| Hopper Hop | $10.2 \pm 1.5$ | $7.8 \pm 2.7$ | $4.0 \pm 0.1$ | $23.2 \pm 4.9$ | Disassemble | $47.6 \pm 5.8$ | $32.4 \pm 5.1$ | $64.4 \pm 12.4$ | $70.4 \pm 8.9$ |
| Walker Walk | $10.3 \pm 3.8$ | $8.30 \pm 1.6$ | $16.7 \pm 4.6$ | $30.5 \pm 6.2$ | Hand Insert | $18.8 \pm 4.0$ | $10.4 \pm 5.6$ | $31.8 \pm 6.21$ | $35.5 \pm 2.3$ |
| Humanoid Walk | $7.6 \pm 3.4$ | $3.2 \pm 0.5$ | $2.6 \pm 0.4$ | $30.1 \pm 7.5$ | Peg Insert Side | $25.3 \pm 10.4$ | $28.9 \pm 5.4$ | $35.0 \pm 3.95$ | $48.2 \pm 3.6$ |
| Dog Trot | $20.5 \pm 12.4$ | $32.9 \pm 6.0$ | $46.6 \pm 4.3$ | $73.5 \pm 6.4$ | Pick Out Of Hole | $28.4 \pm 5.7$ | $42.3 \pm 9.7$ | $42.5 \pm 6.4$ | $66.3 \pm 7.2$ |
| Cup Catch | $60.2 \pm 10.3$ | $56.7 \pm 8.9$ | $93.7 \pm 1.8$ | $89.2 \pm 13.2$ | Pick Place Wall | $30.7 \pm 8.5$ | $42.5 \pm 10.9$ | $58.1 \pm 16.7$ | $63.2 \pm 9.8$ |
| Reacher Hard | $33.9 \pm 9.2$ | $40.7 \pm 8.5$ | $42.3 \pm 5.6$ | $64.9 \pm 5.8$ | Shelf Place | $19.5 \pm 6.4$ | $21.2 \pm 8.3$ | $18.7 \pm 5.15$ | $32.4 \pm 6.5$ |
| Cheetah Run | $26.7 \pm 3.8$ | $27.3 \pm 4.4$ | $33.1 \pm 4.8$ | $39.5 \pm 9.7$ | Stick Pull | $30.2 \pm 4.6$ | $28.5 \pm 9.6$ | $45.6 \pm 17.3$ | $52.4 \pm 5.6$ |
| Quadruped Walk | $15.6 \pm 9.0$ | $14.5 \pm 7.2$ | $18.2 \pm 4.9$ | $63.2 \pm 4.0$ | | | | | |
| Quadruped Run | $40.6 \pm 6.7$ | $43.2 \pm 4.2$ | $64.0 \pm 2.4$ | $61.3 \pm 8.5$ | | | | | |

**Table 3:** Few-shot results for pretrained visual representations (Parisi et al., 2022; Xiao et al., 2022; Nair et al., 2022; Majumdar et al., 2023)

## C   IMPLEMENTATION DETAILS

**Dataset** For six pretraining tasks of Deepmind Control Suite, we train visual RL agents for individual tasks with DrQ-v2 Yarats et al. (2022) until convergence, and we store all the historical interaction steps in a separate buffer. Then, we sample 200 trajectories from the buffer for all tasks except for Humanoid Stand and Dog Walk. Since these two tasks are significantly harder, we use 1000 pretraining trajectories instead. Each episode in Deepmind Control Suite consists of 500 time steps. In terms of the randomly collected dataset, we sample trajectories by taking actions with each dimension independently sampled from a uniform distribution $\mathcal{U}(-1., 1.)$ For MetaWorld, we collect 1000 trajectories for each task, where each episode consists of 200 time steps. We add a Gaussian noise of standard deviation 0.3 to the provided scripted policy.

**Pretraining** For the shallow convolutional network, we follow the same architecture as in Yarats et al. (2022) and add a layer normalization on top of the output of the ConvNet encoder. We set the feature dimension of the ConNet encoder to be 100. In total, this encoder has around 3.95 million parameters.

```python
class Encoder(nn.Module):
    def __init__(self):
        super().__init__()
        self.repr_dim = 32 * 35 * 35

        self.convnet = nn.Sequential(nn.Conv2d(84, 32, 3, stride=2),
                          nn.ReLU(), nn.Conv2d(32, 32, 3, stride=1),
                          nn.ReLU(), nn.Conv2d(32, 32, 3, stride=1),
                          nn.ReLU(), nn.Conv2d(32, 32, 3, stride=1),
                          nn.ReLU())
        self.trunk = nn.Sequential(nn.Linear(self.repr_dim, feature_dim),
                          nn.LayerNorm(feature_dim), nn.Tanh())

    def forward(self, obs):
        obs = obs / 255.0 - 0.5
        h = self.convnet(obs).view(h.shape[0], -1)
        return self.trunk(h)
```

**Listing 1:** Shallow Convolutional Network Architecture Used in `Premier-TACO`

For `Premier-TACO` loss, the number of timesteps $K$ is set to be 3 throughout the experiments, and the window size $W$ is fixed to be 5. Action Encoder is a two-layer MLP with input size being the action space dimensionality, hidden size being 64, and output size being the same as the dimensionality of action space. The projection layer $G$ is a two-layer MLP with input size being feature dimension plus the number of timesteps times the dimensionality of the action space. Its hidden size is 1024. In terms of the projection layer $H$, it is also a two-layer MLP with input and output size both being the feature dimension and hidden size being 1024. Throughout the experiments, we set the batch size to be 4096 and the learning rate to be 1e-4. For the contrastive/self-supervised based baselines, CURL, ATC, and SPR, we use the same batch size of 4096 as `Premier-TACO`. For Multitask TD3+BC and Inverse dynamics modeling baselines, we use a batch size of 1024.

**Imitation Learning** A batch size of 128 and a learning rate of 1e-4 are used. During behavior cloning, we finetune the Shallow ConvNet Encoder. However, when we applied `Premier-TACO` for the large pre-trained ResNet/ViT model, we keep the model weights frozen.

In total, we take 100,000 gradient steps and conduct evaluations for every 1000 steps. For evaluations within the DeepMind Control Suite, we utilize the trained policy to execute 20 episodes, subsequently recording the mean episode reward. In the case of MetaWorld, we execute 50 episodes and document the success rate of the trained policy. We report the average of the highest three episode rewards/success rates from the 100 evaluated checkpoints.

**Computational Resources** For our experiments, we use 8 NVIDIA RTX A6000 with PyTorch Distributed DataParallel for pretraining visual representations, and we use NVIDIA RTX2080Ti for downstream imitation learning.

## D AN ADDITIONAL ABLATION STUDY ON NEGATIVE EXAMPLE SAMPLING STRATEGY

In `Premier-TACO`, we sample one negative example from a size $W$ window centered at the positive example for each data point. However, in principle, we could also use all samples within this window as negative examples instead of sampling only one. In the table below, we compare the performance of two negative example sampling strategies across 10 unseen Deepmind Control Suite tasks. **Bold numbers** indicate the better results.

| | Sampling 1 | Sampling All |
|---|---|---|
| Finger Spin | **75.2 $\pm$ 0.6** | 70.2 $\pm$ 8.4 |
| Hopper Hop | 75.3 $\pm$ 4.6 | **76.1 $\pm$ 3.0** |
| Walker Walk | 88.0 $\pm$ 0.8 | **88.5 $\pm$ 0.4** |
| Humanoid Walk | 51.4 $\pm$ 4.9 | **56.4 $\pm$ 8.9** |
| Dog Trot | **93.9 $\pm$ 5.4** | 92.1 $\pm$ 4.0 |
| Cup Catch | **98.9 $\pm$ 0.1** | 98.3 $\pm$ 1.6 |
| Reacher Hard | **81.3 $\pm$ 1.8** | 80.1 $\pm$ 5.8 |
| Cheetah Run | 65.7 $\pm$ 1.1 | **69.3 $\pm$ 2.3** |
| Quadruped Walk | 83.2 $\pm$ 5.7 | **85.4 $\pm$ 4.2** |
| Quadruped Run | 76.8 $\pm$ 7.5 | **82.1 $\pm$ 9.1** |
| Overall | 79.0 | **79.8** |

**Table 4:** Results of two different negative sampling strategies on 10 unseen Deepmind Control Suite Tasks.

As shown in Table 4, we find that using all samples from the size $W$ window does not significantly enhance performance compared to `Premier-TACO`. Moreover, this approach considerably increases the computational overhead. Given these results, we chose a more computationally efficient strategy of sampling a single negative example from the size $W$ window in `Premier-TACO`.

