# OpenReview forum: "$\texttt{PREMIER-TACO}$ is a Few-Shot Policy Learner: Pretraining Multitask Representation via Temporal Action-Driven Contrastive Loss"
_ICLR.cc/2024/Conference — Submitted to ICLR 2024_

### Official Review · Reviewer_YiPn · 2023-10-30

**Soundness:** 3 good
**Presentation:** 3 good
**Contribution:** 2 fair
**Rating:** 5
**Confidence:** 3

**Summary:**

The paper aims to learn representations for sequential decision-making tasks. Based on temporal action contrastive learning (TACO), the authors adopt a negative sampling strategy to improve the representation especially in multitask contexts. Employing a shallow ConvNet, the authors benchmark their method on Deepmind Control Suite and MetaWorld.

**Strengths:**

1. The paper is well-motivated with careful discussions on the challenges and criteria for learning decision-making representations, as well as the shortage of baseline TACO.
2. The introduced method, Premier-TACO, seems easy to implement.
3. The results indicate the effectiveness of the proposed method.

**Weaknesses:**

1. The introduced Premier-TACO shows incremental contribution over the baseline TACO. Specifically, the only difference is the contrastive loss adopted by TACO and the triplet loss adopted by Premier-TACO. The technique of negative sampling is quite common in the area of metric learning and widely adopted in applications other than robotics, e.g., face recognition. Moreover, the empirical comparison with the baseline TACO is very limited in this paper. The effectiveness of simply adding negative sampling is questionable.

2. The authors repeated several times the infeasibility of adopting visual foundation models (such as those trained on ImageNet or Ego4D) in sequential decision-making tasks. However, it is ungrounded. The authors should evaluate these models (e.g., CLIP, DINOv2, EgoVLP, etc.) on the same benchmark for comparison.

3. The model adopted in the paper is quite small. The pretraining is only performed on synthetic data and also small-scale. The practicality of the method needs further study.

**Questions:**

Why TACO in Fig.9 achieves the same results with different batch sizes?

---

> ### Author Response · Authors · 2023-11-14
> **Response to Reviewer YiPn**
>
> Thanks Reviewer YiPn for the valuable review of our paper. We appreciate the questions you raised and are committed to delivering a comprehensive response to address the issues.
>
> ---
> > 1. The introduced Premier-TACO shows incremental contribution over the baseline TACO. Specifically, the only difference is the contrastive loss adopted by TACO and the triplet loss adopted by Premier-TACO. The technique of negative sampling is quite common in the area of metric learning and widely adopted in applications other than robotics, e.g., face recognition. Moreover, the empirical comparison with the baseline TACO is very limited in this paper. The effectiveness of simply adding negative sampling is questionable.
>
> In response to your suggestion, we have also included additional discussions on existing negative sampling strategies for contrastive learning related to Premier-TACO in Appendix A. We have also updated Table 1 and 2 with TACO as a separate column and updated Figure 9 with a comparison of Premier-TACO vs. TACO in more detail.
>
> First, Premier-TACO is tailored toward a fundamentally different setting than TACO. While TACO is designed for task-specific representation learning in single-task offline and online reinforcement learning, Premier-TACO aims for a broader representation pretraining objective, which is to learn a universal/foundational visual representation from multitask offline datasets. To scale up to multitask offline pretraining, we propose the simple yet highly effective negative sampling strategy in Premier-TACO, which is the key to both scalability and great empirical performance. Without this modification, TACO is not competitive, as shown in the updated Table 1 and 2 as well as Figure 9.
>
> Next, we believe that the general question of multitask representation learning as a foundational model for efficient downstream policy learning itself holds substantial significance for both the machine learning and robotics communities. We acknowledge that our approach bears some resemblance to existing hard negative sampling strategies in contrastive learning and metric learning literature. However, few of them have been successfully applied to the sequential decision making setting, which has to deal with unique challenges such as data distribution shifts between data collection policy and learned policy, task heterogeneity, and pretraining data qualities, as discussed in the introduction.
>
> Furthermore, in our paper, we also discuss the additional aspects of generalization to unseen embodiments and camera views, pretraining dataset quality, and, importantly, how to finetune/adapt existing pre-trained visual encoders for control. These aspects are unique and also crucial to pretraining for sequential decision-making tasks and are under-explored in the existing literature. Many existing works pretrain only on vision datasets and then apply the pretrained vision foundational models directly for downstream policy learning. Thus they often overlook control-relevant considerations and thus suffer from a domain gap between pretraining datasets and downstream visuomotor tasks. Addressing this gap, Premier-TACO proposes a nice solution that could effectively adapt large pre-trained visual encoders for control purposes, as illustrated in Figure 8. Therefore, we firmly believe that this adds significant value to our contribution.
>
> > 2. The authors repeated several times the infeasibility of adopting visual foundation models (such as those trained on ImageNet or Ego4D) in sequential decision-making tasks. However, it is ungrounded. The authors should evaluate these models (e.g., CLIP, DINOv2, EgoVLP, etc.) on the same benchmark for comparison.
>
> We would like to kindly point out that in the main experimental results of Table 1 and Table 2, we have already compared with the **best** pretrained visual encoders (PVRs) / vision foundational models across 4 existing works on applying PVRs for control tasks. (We present the best PVR results in Table 1, and the results of all four PVRs are in Table 3 of Appendix B.2.)
> With a significant performance gain across 18 unseen tasks and 2 different domains, we show that existing works on adapting visual foundational models directly to downstream control tasks with large domain gaps without domain-specific finetuning are suboptimal.

---

> > ### Author Response · Authors · 2023-11-14
> > **Response to Reviewer YiPn**
> >
> > > 3. The model adopted in the paper is quite small. The pretraining is only performed on synthetic data and also small-scale. The practicality of the method needs further study.
> >
> >
> > In terms of model size, although in Tables 1 and 2 the results are shown with a shallow CNN, we would like to emphasize that in Figure 8, we also finetune the large pretrained vision foundational models (R3M with ResNet as its backbone) on domain-specific datasets with Premier-TACO objective. This demonstrates the compatibility and scalability of our approach to larger visual foundational models.
> >
> > Regarding the scope of our experiments, we have conducted pretraining on 6 tasks from the Deepmind Control Suite and 10 tasks in Metaworld. We evaluated the performance of our pretrained representation on 18 unseen tasks in total across these two domains against a comprehensive list of baseline methods. Additionally, we have rigorously tested the effects of varying dataset quality and camera views. This comprehensive evaluation should sufficiently demonstrate the versatility, efficiency, and robustness of our method, addressing the concerns about scale and practicality.
> >
> > We acknowledge the reviewer's suggestion of applying our Premier-TACO method to real robotic datasets, such as RoboNet or Bridge-v2, to further validate its practical applicability. While we agree that this would be an ideal next step, it falls outside the scope of the current paper. We appreciate your feedback and incorporate this point into the discussion of our conclusion section.
> >
> > > Why TACO in Fig.9 achieves the same results with different batch sizes?
> >
> > As we explained in the second line of Page 9 from our original manuscript, in Figure 9, we compared with TACO using a batch size of 4096, the same batch size as used in the main experimental results (Table 1 and 2). But to provide a finer empirical comparison with TACO, we now also update Figure 9 and plot the few-shot imitation learning performance of TACO across different batch sizes.

---

> > > ### Author Response · Authors · 2023-11-21
> > > **Any additional questions?**
> > >
> > > Dear Reviewer YiPn,
> > >
> > > In our earlier response and revised manuscript, we have conducted additional experiments and provided detailed clarifications based on your questions and concerns. As we are ending the stage of the author-reviewer discussion soon, we kindly ask you to review our revised paper and our response and consider adjusting the scores if our response has addressed all your concerns. Otherwise, please let us know if there are any other questions. We would be more than happy to answer any further questions.

---

> ### Author Response · Authors · 2023-11-22
>
> Dear Reviewer YiPn,
>
> Since today is the last day of the author-reviewer discussion period, we kindly ask you to review our previous response and revised manuscript and let us know if there are any other questions. We are happy to answer any additional questions you may have. Thank you.

---

### Official Review · Reviewer_7fxb · 2023-10-31

**Soundness:** 3 good
**Presentation:** 3 good
**Contribution:** 2 fair
**Rating:** 5
**Confidence:** 3

**Summary:**

This paper proposes Premier-TACO, a few-shot policy learner for sequential decision-making tasks. This method is build upon the existing work, i.e., temporal action contrastive learning (TACO) objective, and employs a negative example sampling strategy, which is beneficial for large-scale multitask offline pretraining. Experiments on Deepmind Control Suite and MetaWorld show superior performance.

**Strengths:**

1. The performance on both seen and unseen tasks are superior than other methods.
2. The proposed one negative sample selection is reasonable since it is harder when selecting the negative sample among a window slot than selecting from a batch as in TACO.

**Weaknesses:**

1. It would be better to add an ablation of using the negative sample strategy in TACO on sequential decision-making tasks. It is important to show the effectiveness of the proposed negative sample selection strategy.
2. Premier-TACO uses additional negative samples selected from a temporal window. Compared with TACO, is the batch size doubled? If it is true, what is the result when decreasing the batch size of Premier-TACO to 1/2$N$ compared with TACO with $N$.
3. Does the selection number influence the performance? How about select more than one samples as negatives?
4. Similarly, the negative sample is selected randomly from $W$ window. How about select the hardest one or easiest one from $W$ window?
5. Is Premier-TACO model-free? If yes, can it be applied on other model structures used in the previous methods, e.g., SPR?

**Questions:**

Please refer to the Weakness Section.

---

> ### Author Response · Authors · 2023-11-14
> **Response Reviewer 7fxb**
>
> Thank you for the valuable review of our paper. We appreciate the questions you raised and are committed to delivering a comprehensive response to address the issues.
>
> ---
> > 1. It would be better to add an ablation of using the negative sample strategy in TACO on sequential decision-making tasks. It is important to show the effectiveness of the proposed negative sample selection strategy.
>
> Thanks for your constructive feedback. we have made updates to Figure 9, presenting a comparison of the averaged performance across 10 unseen tasks from Deepmind Control Suite between Premier-TACO (with different batch sizes) and TACO. Additionally, we have revised Table 1 and Table 2 by incorporating an additional column for TACO in the baselines, facilitating a direct comparison with Premier-TACO, as per your suggestion. These modifications to Table 1 and 2 as well as the updated Figure 9 effectively illustrate the efficacy of the novel negative sample selection strategy in Premier-TACO, where Premier-TACO outperforms TACO with at least 40\% across all batch sizes. We believe that these additions should address your concern on the empirical comparison against TACO baseline and we welcome any further discussion or suggestions from the reviewer.
>
>
>
> ---
> > 2. Premier-TACO uses additional negative samples selected from a temporal window. Compared with TACO, is the batch size doubled? If it is true, what is the result when decreasing the batch size of Premier-TACO to 1/2N compared with TACO with N?
>
> Thank you for your question regarding the batch size used in Premier-TACO. To clarify, the batch size in Premier-TACO is not doubled compared to TACO. As outlined on page 9, line 2 of our manuscript, and demonstrated in Figure 9, we compare Premier-TACO and TACO using an fixed batch size of 4096. This batch size is consistent with that used in our main experimental results (Tables 1 and 2).
>
> To address your question further, we have updated Figure 9 to include comparisons of Premier-TACO and TACO both at different batch sizes. This additional analysis should provide a clearer understanding of how batch size variations impact the performance of both methods.
>
> Finally, in the update to Table 1, we have included a column for TACO where it is pretrained with the same batch size of 4096, aligning it with other contrastive learning baselines. This ensures that the comparisons made are fair and consistent across different methods.
>
>
> > 3. Does the selection number influence the performance? How about select more than one samples as negatives?
>
> Thank you for this insightful question. Indeed, we have considered this in the original design of our learning objective. In the table below, we compare the performance when selecting a single negative sample randomly from a size W window against when using all samples within this window across 10 unseen Deepmind Control Suite tasks.
>
> |                | Sampling 1     | Sampling All|
> | -----------    | ----     | ----------- |
> | Finger Spin    | 75.2 ± 0.6 | 70.2 ± 8.4 |
> | Hopper Hop     | 75.3 ± 4.6 | 76.1 ± 3.0 |
> | Walker Walk    | 88.0 ± 0.8 | 88.5 ± 0.4 |
> | Humanoid Walk  | 51.4 ± 4.9 | 56.4 ± 8.9 |
> | Dog Trot       | 93.9 ± 5.4 | 92.1 ± 4.0 |
> | Cup Catch      | 98.9 ± 0.1 | 98.3 ± 1.6 |
> | Reacher Hard   | 81.3 ± 1.8 | 80.1 ± 5.8 |
> | Cheetah Run    | 65.7 ± 1.1 | 69.3 ± 2.3 |
> | Quadruped Walk | 83.2 ± 5.7 | 85.4 ± 4.  |
> | Quadruped Run  | 76.8 ± 7.5 | 82.1 ± 9.1 |
> | Overall        |    79.0    |  79.8 |
>
>
> Our findings indicate that using all samples from the size W window does not significantly enhance performance compared to Premier-TACO. Moreover, this approach considerably increases the computational overhead. Given these results, we chose a more computationally efficient strategy of sampling a single negative example from the size W window. We appreciate this question that you raised, and we have incorporated it into Appendix D of the updated manuscript.

---

> > ### Author Response · Authors · 2023-11-14
> > **Response to Reviewer 7fxb**
> >
> > ---
> > >4. Similarly, the negative sample is selected randomly from W window. How about select the hardest one or easiest one from W window?
> >
> > This aspect is addressed in our ablation study on window size, detailed in Figure 10. The selection of the hardest negative example, which would be adjacent to the positive sample, is effectively the same as setting the window size W to 1. Conversely, choosing the easiest negative sample is akin to expanding the window size. Our results show that selecting negatives that are either too challenging (W=1) or too simple (W=9) adversely affects the overall objective of representation learning. This finding guided our decision to balance the difficulty of the negative examples.
> >
> > ---
> > > 5. Is Premier-TACO model-free? If yes, can it be applied on other model structures used in the previous methods, e.g., SPR?
> >
> >
> > Premier-TACO is not itself strictly a model-free or a model-based RL algorithm. Instead, it focuses on pretraining a versatile feature representation capable of capturing essential environmental dynamics from a multitask offline dataset and is applied to few-shot imitation learning. In our experiments, we provide a detailed comparison between Premier-TACO, SPR, and other contrastive learning methods in Tables 1 and 2. It's important to note that Premier-TACO employs a distinct approach for representation learning compared to SPR and other existing methods.
> >
> > Furthermore, Premier-TACO exhibits flexibility in network architectures; we use a shallow ConvNet encoder in Table 1 and 2, but our method can be applied to finetune models such as R3M that employ more complex architectures such as ResNet, as demonstrated in Figure 8. We hope this clarification is helpful. If you have any further questions or suggestions, we are more than happy to answer them.

---

> > > ### Author Response · Authors · 2023-11-21
> > > **Any additional questions?**
> > >
> > > Dear Reviewer 7fxb,
> > >
> > > In our earlier response and revised manuscript, we have conducted additional experiments and provided detailed clarifications based on your questions and concerns. As we are ending the stage of the author-reviewer discussion soon, we kindly ask you to review our revised paper and our response and consider adjusting the scores if our response has addressed all your concerns. Otherwise, please let us know if there are any other questions. We would be more than happy to answer any further questions.

---

> ### Author Response · Authors · 2023-11-22
>
> Dear Reviewer 7fxb,
>
> Since today is the last day of the author-reviewer discussion period, we kindly ask you to review our previous response and revised manuscript and let us know if there are any other questions. We are happy to answer any additional questions you may have. Thank you.

---

### Official Review · Reviewer_AXkc · 2023-11-06

**Soundness:** 3 good
**Presentation:** 3 good
**Contribution:** 2 fair
**Rating:** 5
**Confidence:** 2

**Summary:**

This paper proposes Premier-TACO, a multitask feature representation learning method, aiming to enhance the efficiency of few-shot policy learning in sequential decision-making tasks. Premier-TACO pretrains a general feature representation using s small subset of multitask offline datasets and then fine-tunes the network to specific tasks with a few experts. Additionally, Premier-TACO employs a negative example sampling strategy on contrastive learning objectives. Experimental results show that Premier-TACO can outperform the state-of-the-art on DeepMind Control Suite and MetaWorld.

**Strengths:**

1.	The paper is well-written and easy to follow.
2.	The proposed method Premier-TACO can simultaneously achieve versatility, efficiency, robustness, and compatibility.
3.	Empirical results demonstrate that Premier-TACO can achieve SOTA results on several benchmarks.

**Weaknesses:**

1.	The novelty is somewhat limited. The proposed method is built on the temporal action constrastive learning (TACO) objective [1]. The overall framework is similar to [1]. The authors additionally employ a negative example sampling strategy. But the negative sampling has been widely used in constrastive learning [2][3]. Considering the above factors, I think that the innovation of the method is limited
2.	The detailed experimental comparisons and discussions with TACO are missed.
3.	In the ablation study, in order to show the effectiveness of the proposed negative example sampling strategy, the authors should compare Premier-TACO with a baseline without using a negative example sampling strategy. The related experimental results should be added.

[1] TACO: Temporal latent action-driven contrastive loss for visual reinforcement learning, NeurIPS 2023.

[2] Robust Contrastive Learning Using Negative Samples with Diminished Semantics, NeurIPS 2021.

[3] Hard Negative Sampling Strategies for Contrastive Representation Learning, arxiv 2022.

**Questions:**

See Weakness for detail.

---

> ### Author Response · Authors · 2023-11-14
> **Response to Reviewer AXkc**
>
> Thanks Reviewer AXKc for the valuable review of our paper. We appreciate the questions you raised and are committed to delivering a comprehensive response to address the issues.
>
> ---
> > 1. The novelty is somewhat limited. The proposed method is built on the temporal action constrastive learning (TACO) objective [1]. The overall framework is similar to [1]. The authors additionally employ a negative example sampling strategy. But the negative sampling has been widely used in constrastive learning [2][3]. Considering the above factors, I think that the innovation of the method is limited.
>
> In response to your suggestion, we have also included additional discussions on existing negative sampling strategies for contrastive learning related to Premier-TACO in the Appendix A. Here, we would like to highlight several key contributions of our work that distinguish our method from TACO [1] and the existing works in hard negative mining  [2,3].
>
> First, Premier-TACO is tailored toward a fundamentally different setting than TACO. While TACO is designed for task-specific representation learning in single-task offline and online reinforcement learning, Premier-TACO aims for a broader representation pretraining objective, which is to learn a universal/foundational visual representation from multitask offline datasets. To scale up to multitask offline pretraining, we propose the simple yet highly effective negative sampling strategy in Premier-TACO, which is the key to both scalability and great empirical performance.
>
> Next, we believe that the general question of multitask representation learning as a foundational model for efficient downstream policy learning itself is holds substantial significance for both the machine learning and robotics communities. In this paper, we show that with the negative example strategy employed by Premier-TACO, it achieves an impressive performance gain compared with a comprehensive set of baseline pretraining methods across two domains. We acknowledge that our approach bears some resemblance to existing hard negative sampling strategies in contrastive learning literature. However, few of them have been successfully applied to the sequential decision making setting, which has to deal with unique challenges such as data distribution shifts between data collection policy and learned policy, task heterogeneity, and pretraining data qualities, as discussed in the introduction.
>
> Furthermore, in our paper, we also discuss the additional aspects of generalization to unseen embodiments and camera views, pretraining dataset quality, and, importantly, how to finetune/adapt existing pre-trained visual encoders for control. These aspects are unique and also crucial to pretraining for sequential decision-making tasks and are under-explored in the existing literature. Many existing works pretrain only on vision datasets and then apply the pretrained vision foundational models directly for downstream policy learning. Thus they often overlook control-relevant considerations and thus suffer from a domain gap between pretraining datasets and downstream visuomotor tasks. Addressing this gap, Premier-TACO proposes a nice solution that could effectively adapt large pre-trained visual encoders for control purposes, as illustrated in Figure 8. Therefore, we firmly believe that this adds significant value to our contribution.
>
> > 2. The detailed experimental comparisons and discussions with TACO are missed.
>
> Thank you for your feedback regarding the comparison with the baseline TACO. In Figure 9 of the original manuscript, we have already presented a comparison of the averaged performance across 10 unseen tasks from the Deepmind Control Suite between Premier-TACO (with different batch sizes) and TACO. We appreciate your suggestion to highlight this comparison in the paper, and so we have updated Table 1 accordingly. This revision includes an additional column for TACO in the baselines, enhancing the clarity of our empirical comparison. For the results of MetaWorld in Table 2, the experiments are still running. We will incorporate the results into our manuscript as soon as they are available.
>
> ====== Update: results of TACO for MetaWorld (Table 2) are now updated ===========

---

> ### Author Response · Authors · 2023-11-14
> **Response to Reviewer AXkc**
>
> > 3. In the ablation study, in order to show the effectiveness of the proposed negative example sampling strategy, the authors should compare Premier-TACO with a baseline without using a negative example sampling strategy. The related experimental results should be added.
>
> Thank you for your suggestion on the additional empirical comparison. In our evaluation now, we have already compared with SPR, which applies a BYOL-based objective that does not leverage negative examples. Additionally, we have also compared with TACO, which uses a naive negative example sampling strategy and treats every other data point in the batch as negative examples. Furthermore, Figure 9 of our ablation study is now updated to include a more fine-grained comparison between Premier-TACO and TACO with different batch sizes. With a significant performance gain shown by Premier-TACO, the results should be sufficient to demonstrate the effectiveness of the negative sampling strategy proposed in Premier-TACO. If there are specific additional baselines that don't use negative sampling that you believe are important to compare against, please let us know.

---

> ### Author Response · Authors · 2023-11-21
> **Any additional questions?**
>
> Dear Reviewer AXkc,
>
> In our earlier response and revised manuscript, we have conducted additional experiments and provided detailed clarifications based on your questions and concerns. As we are ending the stage of the author-reviewer discussion soon, we kindly ask you to review our revised paper and our response and consider adjusting the scores if our response has addressed all your concerns. Otherwise, please let us know if there are any other questions. We would be more than happy to answer any further questions.

---

> ### Author Response · Authors · 2023-11-22
>
> Dear Reviewer AXkc,
>
> Since today is the last day of the author-reviewer discussion period, we kindly ask you to review our previous response and revised manuscript and let us know if there are any other questions. We are happy to answer any additional questions you may have. Thank you.

---

### Author Response · Authors · 2023-11-14

We thank all reviewers for their insightful questions and valuable feedback. We have addressed all reviewers' individual questions in separate responses. Here, we briefly outline the major updates to the revised submission for reviewers' reference.

**Writing**: We incorporate all of the reviewers' suggestions on paper writing. In particular, we have added a detailed discussion on existing works for hard negative example sampling strategies in Appendix A (AXKc, YiPn). We have also added a discussion on pretraining for real robotics datasets as the future direction of our work (YiPn).

**Experiments**: We have added a complete comparison with TACO as a separate column in Table 1 that compares Premier-TACO with other pretraining method on the Deepmind Control Suite (AXKc, 7fxb, YiPn). Additionally, we also updated Figure 9 with TACO using different pretraining batch sizes to provide a better empirical comparison of Premier-TACO versus TACO. Finally, we incorporated the suggestion of 7fxb to add an additional ablation study on the negative sampling strategy of Premier-TACO (sampling one vs. sampling all from the window). The finding indicates that, compared to sampling all items from the window as negatives, sampling just one negative in Premier-TACO achieves a similar final performance but is considerably more computationally efficient.

Since all reviewers have questions regarding the novelty of Premier-TACO compared with TACO and other related works in contrastive learning, here we would like to clarify again the novelty/contribution of Premier-TACO.

First, Premier-TACO is designed for a much broader objective than TACO: pretraining shared representation from diverse multitask offline datasets, as opposed to TACO's focus on single-task learning. Also, our work is distinct from many other existing works that pretrain only on vision datasets and then apply the pretrained vision foundational models directly for downstream policy learning, which often overlook control-relevant considerations and thus suffer from a domain gap between pre-training datasets and downstream visuomotor tasks. For Premier-TACO, we focus on the question of how to pretrain/finetune a universal visual representation for control with in-domain multitask offline datasets that enable few-shot imitation learning to unseen tasks. We believe that this question holds substantial significance for both the machine learning and robotics communities.


As such, we need to deal with unique challenges in sequential decision making, such as data distribution shifts between data collection policy and learned policy, task heterogeneity, and pretraining data qualities, as discussed in the introduction. Toward this end, Premier-TACO proposes a simple yet highly effective negative sampling strategy. Contrary to TACO, where negative examples could come from a diverse set of tasks with completely different visual appearances and thus be much less effective, Premier-TACO strategically samples one negative example from a nearby window of the positive example. Although straightforward, this modification allows Premier-TACO to easily scale up to multitask offline pretraining without scaling the batch size with the number of pretraining tasks.

Furthermore, the value of Premier-TACO is substantiated through extensive empirical evaluation: without these modifications, TACO is not competitive. Notably, with a limited number of expert demonstrations—fewer than those examined in existing works as detailed in the experiment section—Premier-TACO obtains impressive few-shot adaptation performance across 18 tasks from two distinct domains. Furthermore, Premier-TACO has several unique benefits, including generalizability to novel embodiments and camera views, resilience against low-quality pretraining data, and compatibility with large-scale pretrained vision foundational models. These aspects are all unique and crucial to sequential decision making tasks, which all add value to the broad applicability of Premier-TACO.

We greatly appreciate all reviewers' suggestions. We hope that our paper updates and responses have addressed reviewers' questions and concerns. Please let us know if you have further questions.

---

### Author Response · Authors · 2023-11-17
**Additional Questions?**

Thank you all for your constructive and insightful feedbacks. We have conducted additional experiments and provided clarifications to address all the questions you raised, as we believe they are crucial for enhancing the quality of our paper. If you have any further questions or concerns, we are more than happy to address them.

---

### Meta-Review · Area_Chair_Y63c · 2023-12-16

**Metareview:**

This paper proposes a multitask feature representation learning methodology aiming to enhance the efficiency of few-shot policy learning in sequential decision-making tasks. Initially, this paper received consistent negative reviews. The authors provided strong rebuttals during the discussion period. However, the reviewers didn't actively engage in the discussion, and they didn't change their final ratings.

After carefully reading the paper, reviews, and rebuttals, I decided not to overturn the reviewers' decisions. The reasons are as follows.

- **About novelty**. Novelty is a major concern mentioned by different reviewers. I checked the rebuttal from the authors, and agree that this paper has a fundamentally different setting from the original TACO. However, at least for the technical details, the paper is built on TACO. Therefore, I partially agree with the reviewers on the technical novelty concern.

- **About the writing and organization.** The writing and organization of this paper should be improved. For example, the current method section is too short. The authors didn't provide enough motivations and explanations to highlight the differences between their method and the original TACO. After reading the method section, especially after seeing Figure 3, I feel that the proposed method has similar technical designs to TACO.

I believe it is fine to build a new method based on the existing ones. However, the claims should match your technical novelty. I don't think it is reasonable to highlight the proposed method is fundamentally different from TACO, because these two methods share so many technical similarities.

I appreciate the authors' efforts in experiments and rebuttals. However, I don't think the current version is strong enough to overturn three negative reviews. Therefore, I recommend rejection for now. I think by incorporating the reviewers' suggestions and adjusting the paper writing, this submission can easily be accepted to next venues.

**Justification For Why Not Higher Score:**

As I mentioned in my meta-review, after the rebuttal session, there are still some unsolved concerns. I agree this is a borderline submission. However, I cannot overturn the three negative ratings for the current version.

**Justification For Why Not Lower Score:**

N/A

---

### Decision · Program_Chairs · 2024-01-16

Reject